# Glycosylated extracellular mucin domains protect against SARS-CoV-2 infection at the respiratory surface

**Maitrayee Chatterjee**[1☯], **Liane Z. X. Huang**[1☯], **Anna Z. Mykytyn**[2☯], **Chunyan Wang**[1¤], **Mart M. Lamers**[2], **Bart Westendorp**[3], **Richard W. Wubbolts**[2], **Jos P. M. van Putten**[1], **Berend-Jan Bosch**[1], **Bart L. Haagmans**[2]*, **Karin Strijbis**[1]*

**1** Department of Biomolecular Health Sciences, Division Infectious Diseases and Immunology, Faculty of Veterinary Medicine, Utrecht University, Utrecht, The Netherlands, **2** Viroscience Department, Erasmus Medical Center, Rotterdam, The Netherlands, **3** Department of Biomolecular Health Sciences, Division Cell Biology, Metabolism and Cancer, Faculty of Veterinary Medicine, Utrecht University, Utrecht, The Netherlands

☯ These authors contributed equally to this work.
¤ Current address: Department of Pathogenic Biology, School of Biomedical Sciences, Shandong University, Jinan, Shandong, P.R. China
* b.haagmans@erasmusmc.nl (BLH); K.Strijbis@uu.nl (KS)

**Data Availability Statement:** All relevant data are within the manuscript and its Supporting information files.

## Abstract

Mucins play an essential role in protecting the respiratory tract against microbial infections while also acting as binding sites for bacterial and viral adhesins. The heavily *O*-glycosylated gel-forming mucins MUC5AC and MUC5B eliminate pathogens by mucociliary clearance. Transmembrane mucins MUC1, MUC4, and MUC16 can restrict microbial invasion at the apical surface of the epithelium. In this study, we determined the impact of host mucins and mucin glycans on epithelial entry of SARS-CoV-2. Human lung epithelial Calu-3 cells express the SARS-CoV-2 entry receptor ACE2 and high levels of glycosylated MUC1, but not MUC4 and MUC16, on their cell surface. The *O*-glycan-specific mucinase StcE specifically removed the glycosylated part of the MUC1 extracellular domain while leaving the underlying SEA domain and cytoplasmic tail intact. StcE treatment of Calu-3 cells significantly enhanced infection with SARS-CoV-2 pseudovirus and authentic virus, while removal of terminal mucin glycans sialic acid and fucose from the epithelial surface did not impact viral entry. In Calu-3 cells, the transmembrane mucin MUC1 and ACE2 are located to the apical surface in close proximity and StcE treatment results in enhanced binding of purified spike protein. Both MUC1 and MUC16 are expressed on the surface of human organoid-derived air-liquid interface (ALI) differentiated airway cultures and StcE treatment led to mucin removal and increased levels of SARS-CoV-2 replication. In these cultures, MUC1 was highly expressed in non-ciliated cells while MUC16 was enriched in goblet cells. In conclusion, the glycosylated extracellular domains of different transmembrane mucins might have similar protective functions in different respiratory cell types by restricting SARS-CoV-2 binding and entry.

**Funding:** MC is supported by One Health funding provided by the Faculty of Veterinary Medicine. KS and LZXH are supported by the European Research Council under the European Union's Horizon 2020 research and innovation program (ERC-2019-STG-852452). The funders had no role in study design, data collection and analysis, decision to publish, or preparation of the manuscript.

**Competing interests:** The authors declare no competing interests.

## Author summary

SARS-CoV-2, the virus that has caused the devastating COVID-19 pandemic, causes a range of symptoms in infected individuals, from mild respiratory illness to acute respiratory distress syndrome. A fundamental understanding of host factors influencing viral entry is critical to elucidate SARS-CoV-2–host interactions and identify novel therapeutic targets. In this study, we investigated the role of host mucins and mucin glycans on SARS-CoV-2 entry into the airway epithelial cells. Mucins are a family of high molecular weight *O*-glycosylated proteins that play an essential role in protecting the respiratory tract against viral and bacterial infections. The gel-forming mucins MUC5AC and MUC5B clear pathogens by mucociliary clearance while transmembrane mucins MUC1, MUC4, and MUC16 can restrict or facilitate microbial invasion at the apical surface of the epithelium. The mucin-selective protease StcE specifically cleaves the glycosylated extracellular part of the mucins without perturbing the underlying domains. We show that removal of mucins from the surface of Calu-3 cells with StcE mucinase increases binding of the SARS-CoV-2 spike protein to the epithelial surface and greatly enhances infection. Enhanced viral replication was also significantly increased in primary airway epithelial cultures treated with StcE mucinase. This study demonstrates the important role of glycosylated extracellular mucin domains as a host defense mechanism during SARS-CoV-2 entry. Future efforts should be focused on characterizing the expression and role of specific soluble and transmembrane mucins in different cell types during the different stages of SARS-CoV-2 infection.

## Introduction

The respiratory mucus system protects the respiratory epithelium against invading pathogens. The major components of mucus are heavily *O*-glycosylated mucin glycoproteins. Soluble mucins are secreted by goblet cells and provide mucus threads for mucociliary clearance (MCC) of particles and pathogens. Transmembrane mucins are expressed on the apical membrane and cilia and prevent access to epithelial surface receptors. The major mucins of the respiratory system are soluble mucins MUC5AC and MUC5B, and transmembrane (TM) mucins MUC1, MUC4, and MUC16 [1]. MUC1 and MUC4 are expressed in the upper and lower airway epithelium, whereas MUC16 expression is restricted to the lower airways [2]. The high molecular weight mucin glycoproteins contain domains with extensive *O*-glycan structures that often terminate with charged sialic acids or hydrophobic fucoses that impact their interaction with microbes [3]. The expression and glycosylation profiles of mucins are directly influenced by colonization and invasion by bacteria and viruses and are altered during inflammation of the respiratory tract [4]. Transmembrane mucins form filamentous structures that extend above the apical surface of the epithelium and these mucins consist of a heavily *O*-glycosylated N-terminal extracellular domain (ED), a single transmembrane domain, and a C-terminal cytoplasmic domain (CT) with signaling capacity. In the lung, MUC1 primarily expresses around microvilli and protrudes at least 100 nm from the cell surface whereas MUC4 (~300 nm in size), and the even larger MUC16 are expressed on the surface of cilia [5] and goblet cells [6]. Together, the TM mucins form a barrier that restricts access to the underlying epithelium, act as releasable decoy receptors, and sterically hinders the binding of pathogens to underlying cellular receptors [7]. MUC1 has been most extensively studied and implicated in defense against respiratory infections with *Pseudomonas aeruginosa* [8], respiratory syncytial virus [9] and influenza A virus infection [10]. SARS-CoV-2, the coronavirus that

is responsible for the COVID-19 pandemic, is an enveloped, single-stranded, positive-sense RNA virus that belong to the β coronavirus genus within the coronaviridae family [11,12]. SARS-CoV-2 preferentially utilizes angiotensin-converting enzyme 2 (ACE2) as entry receptor by interaction with its envelope-anchored spike protein [13]. In addition to ACE2, SARS-CoV-2 entry requires proteolytic cleavage of the spike protein that can be mediated by the transmembrane serine protease 2 (TMPRSS2) [14]. Human coronaviruses have also been described to depend on sialic acids linked to glycoproteins or gangliosides as primary attachment sites in the respiratory tract [15]. Glycosylated mucins can be decorated with sialic acids and therefore might provide viral binding sites, or on the other hand form a barrier that restricts access to the ACE2 receptor. In this study, we investigated the role of transmembrane mucins and their terminal glycans during SARS-CoV-2 entry in a respiratory cell line and primary airway cultures. We show that MUC1 is abundantly expressed on the respiratory Calu-3 cell line, and that both MUC1 and MUC16 are present on the surface of organoid-derived air-liquid interface (ALI) differentiated airway cultures. Enzymatic removal of extracellular mucin domains greatly enhances SARS-CoV-2 spike protein binding and viral infection. This study points towards a critical role for transmembrane mucins in limiting SARS-CoV-2 infection.

## Results

### MUC1 is highly expressed on the surface of ACE2-positive respiratory epithelial cells

The Human Cell Atlas consortium respiratory single cell RNA-seq dataset allows analysis of gene expression in the nasal cavity and proximal, intermediate, and distal respiratory tract [16]. We analysed this dataset to determine the expression of ACE2 and mucins in different respiratory cell types present in the nasal (N), and upper and lower respiratory mucosa. ACE2-positive cells included secretory, basal, suprabasal and multiciliated cells. The majority of secretory and multiciliated cells expressed the major TM mucin MUC1 (Fig 1A). Next, we determined the mucin repertoire of ACE2-positive cells in the nasal mucosa and lower respiratory tract. MUC1 was expressed by the majority of cells to a relatively high extent, while TM mucins MUC4 and MUC16 were abundant in multiciliated cells and soluble mucins MUC5AC and MUC5B were highly expressed in secretory and goblet cells (Fig 1B). This analysis suggests that different respiratory cell types have unique mucin repertoires and that the TM mucin MUC1 is the most abundantly expressed mucin in most types of ACE2-positive respiratory cells.

The human respiratory Calu-3 cell line expresses ACE2 and TMPRSS2 and is highly susceptible to SARS-CoV-2 spike protein-mediated entry [14,17,18]. We first determined the expression of different mucins and their glycans on Calu-3 cells by immunofluorescence confocal microscopy. Multiple Z-stack images showed expression of MUC1, MUC4, and MUC5AC but only very limited expression of MUC16 (Fig 1C, S1A Fig respectively). To distinguish which mucins are expressed on the extracellular cell surface, we performed immunofluorescence staining without permeabilization of the Calu-3 cells. Using this method, MUC1 was clearly detectable on the cell surface whereas MUC4 and MUC5AC could not be stained indicating either intracellular localization or limitations of antibodies to detect these two mucins on the cellular surface (Fig 1D). Next, we determined the expression of the terminal mucin glycans sialic acid and fucose on Calu-3 cells. Immunofluorescence with SNA, MALII, and UEAI lectins showed the presence of α-2,6 sialic acid, α-2,3 sialic acid, and fucose on Calu-3 cells, respectively (Fig 1E). The α-2,6 sialic acid and α-2,3 sialic acid signals (SNA and MALII) were more prominently detected at the edge of the cell island compared to the fucose signal (UEAI) and some colocalization with MUC1 could be observed. These results demonstrate that Calu-3

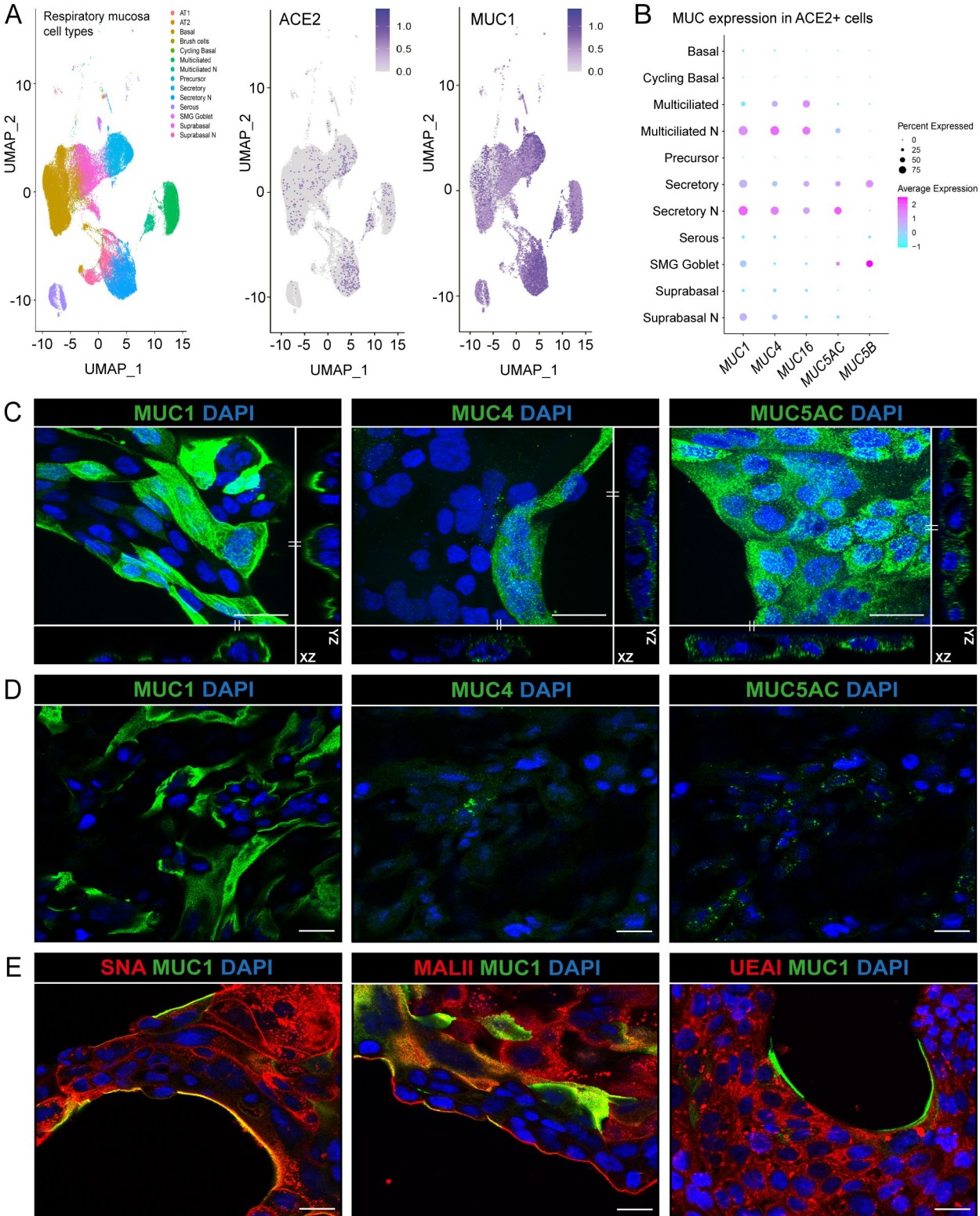

**Fig 1. Expression of mucins in respiratory epithelial cells.** (A) scRNA-seq analysis of ACE2 and MUC1 expression in different cell types in the respiratory mucosa. Dataset include samples from nasal cavity (N), upper, intermediate and lower respiratory tract [16]. (B) Expression of TM mucins MUC1, MUC4 and MUC16 and gel-forming mucins MUC5AC and MUC5B in ACE2 positive cells. MUC1 is the most highly expressed mucin in ACE2-positive cells. (C) Immunofluorescence confocal microscopy images showing expression of TM mucins MUC1 (214D4, green),

MUC4 (8G7, green) and gel-forming mucin MUC5AC (MUC5AC, green) in permeabilized Calu-3 cells. Maximum projections and side views of Z-stacks are shown. (D) Immunofluorescence confocal microscopy without permeabilization showing expression of MUC1 on the surface of Calu-3 cells. MUC4 and MUC5AC could barely be detected suggesting intracellular localization. (E) Immunofluorescence confocal microscopy imaging for α-2,6 sialic acids (SNA, red), α-2,3 sialic acids (MALII, red) and fucose (UEAI, red) in combination with MUC1 (214D4 antibody, green) demonstrates high levels of sialic acid and fucose in Calu-3 cells. Nuclei were stained with DAPI (blue). White scale bars represent 20 μm.

cells endogenously express MUC1 on their surface and have abundant expression of sialic acids and fucose.

## StcE cleaves the MUC1 glycosylated ED and does not affect ACE2 expression

The StcE mucinase recognizes an *O*-glycosylated serine-threonine motif that is abundant in mucins and is virtually absent in non-mucin proteins and results in cleavage of the mucin polypeptide backbone at the recognition site [19]. We previously applied this bacterial mucinase and its inactive point mutant E447D to remove the MUC1 ED [20]. To investigate the effect of StcE on endogenous MUC1 expressed by Calu-3 cells, confocal microscopy was performed on non-treated, StcE-treated and E447D-treated Calu-3 cells and stained with α-MUC1-ED antibody 214D4, α-MUC1-SEA antibody 232A1, and α-MUC1-CT antibody CT2. StcE treatment resulted in efficient cleavage and removal of the MUC1 glycosylated domain as indicated by a complete loss of α-MUC1-ED 214D4 staining after incubation with the enzyme (Fig 2A). The MUC1 SEA domain and CT are predicted not to be digested by StcE and indeed both domains remained detectable after enzyme treatment (Fig 2B and 2C). We next investigated the effect of StcE, E447D, neuraminidase, and fucosidase treatment on MUC1 by Western blot. Calu-3 cells were incubated with the enzymes for 3 h and then subjected to Western blot analysis with the α-MUC1-ED antibody 214D4 and α-MUC1-CT antibody CT2. After incubation with StcE, the glycosylated part of the extracellular domain of MUC1 (about 450 kDa) was no longer detectable. The high molecular weight MUC1 band was not affected by treatment with the inactive enzyme E447D or fucosidase. After neuraminidase treatment some reduction of the MUC1 signal compared to the loading control was observed (Fig 2D and 2E). In a dotblot analysis this difference was less apparent, perhaps pointing at reduced transfer from gel to blot of mucins without negatively charged sialic acids (Fig 2D and 2E). The observed banding pattern for the MUC1 cytoplasmic tail was not affected by the enzymatic treatments (Fig 2F). Furthermore, we wanted to determine the effect of enzymatic treatment on ACE2 stability because the ACE2 receptor itself is glycosylated [21]. No change in expression of the full-length glycosylated ACE2 (nearly 140 kDa) could be observed after treatment with StcE, E447D, neuraminidase, or fucosidase. Interestingly, the soluble form of ACE2 (around 70 kDa) was more prominently detectable after fucosidase treatment (Fig 2G). These results demonstrate that StcE cleaves the glycosylated part of the MUC1 ED without affecting ACE2 expression in Calu-3 cells. To investigate the effect of StcE treatment on *O*-glycosylated surface proteins other than MUC1, we stained the treated and untreated Calu-3 cells with a fluorescently labelled mucin binding domain derived from StcE (X409-GFP) [22]. Because StcE treatment only removes surface mucins, confocal analysis was performed on non-permeabilized Calu-3 cells to detect only surface mucins and on permeabilized Calu-3 cells to stain for both surface and intracellular mucins. A limited punctate staining was observed for X409 on the Calu-3 surface while MUC1 ED staining showed a more continuous surface staining as previously observed. The MUC1 signal on the cellular surface was completely lost after StcE treatment, while some staining remained for X409 (Fig 2H). With permeabilized cells, the MUC1 signal was again completely lost after StcE treatment but a higher level of remaining

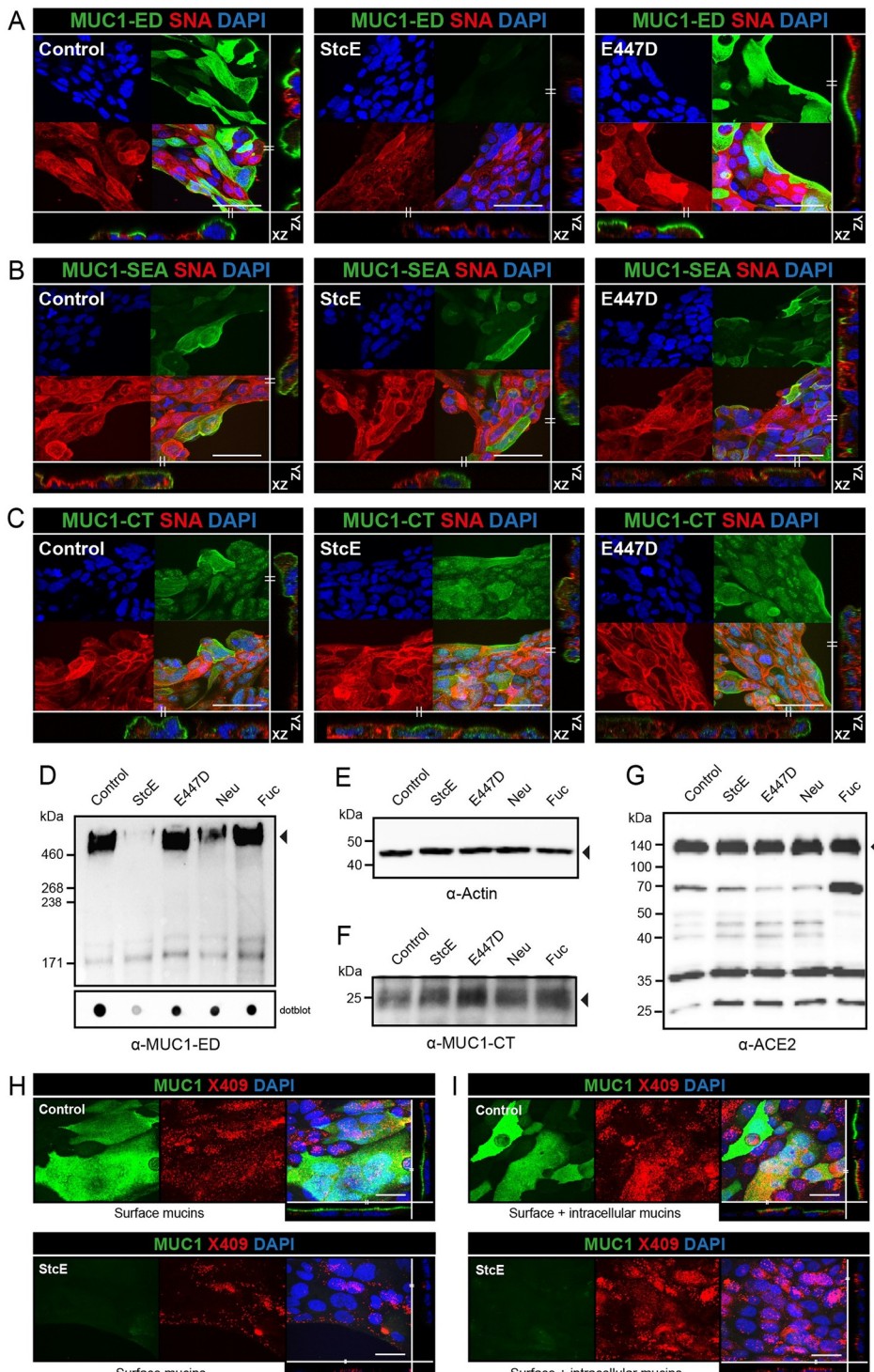

**Fig 2. StcE specifically cleaves the glycosylated MUC1 ED and does not affect ACE2 expression.** (A) Immunofluorescence confocal microscopy images showing Calu-3 cells treated with StcE or E447D stained for the glycosylated part of the MUC1 extracellular domain (214D4, green) and α-2,6-linked sialic acids (SNA, red). Complete loss of 214D4 signal was observed after treatment with StcE. (B,C) Immunofluorescence confocal microscopy images of Calu-3 cells as above stained for the MUC1 SEA domain (α-MUC1-SEA antibody 232A1, green) or cytoplasmic tail of MUC1 (α-MUC1-CT antibody CT2, green) in combination with α-2,6-linked sialic acids (SNA, red). The SEA domain and CT were not affected by StcE treatment. Nuclei were stained with DAPI (blue). White scale bars represent 20 μm. Western blot and dotblot analysis of 7-day grown Calu-3 cells incubated with indicated enzymes for 3 h at 37°C

stained with α-MUC1-ED antibody 214D4 (D), β-actin loading control (E), the MUC1 cytoplasmic tail with α-MUC1 CT antibody CT2 (F), and ACE2 (G). StcE treatment removes the MUC1 ED but does not affect the MUC1 CT or ACE2 receptor. (H) Immunofluorescence confocal microscopy images showing Calu-3 cells treated with StcE stained for the glycosylated part of the MUC1 extracellular domain (214D4, green) and fluorescently labelled mucin binding domain derived from StcE (X409-GFP) (X409, red). More continuous surface staining for MUC1 ED and limited punctate staining with X409 on the non-permeabilized cells Calu-3 cells. The MUC1 signal was completely removed after StcE treatment, while some staining remained for X409. (I) Immunofluorescence confocal microscopy images showing, a comparable result with a higher level of remaining X409 signal with permeabilized cells. White scale bars represent 20 μm.

intracellular X409-reactive mucin-like proteins was observed (Fig 2I). These results reinforce that MUC1 is highly expressed on the surface of Calu-3 cells, but also demonstrate the presence of some other *O*-glycosylated mucin(-like) proteins.

## Enzymatic removal of mucin domains enhances SARS-CoV-2 entry into Calu-3 cells

Next, we investigated whether the sialic acid and fucose residues on glycans or complete glycosylated mucin domain can impact SARS-CoV-2 infection of respiratory cells. Surface α2,3-, α2,6-, and α2,8-linked sialic acids were removed by incubation with neuraminidase and fucose by fucosidase treatment and StcE was used to remove mucin extracellular domains. Neuraminidase treatment for 3 h removed the majority of surface-exposed α2,3- linked sialic acids as detected by MALII staining, and α2,6-linked sialic acids detected by SNA staining (S1B Fig). Similarly, fucosidase treatment for 3 h cleaved surface-exposed fucose which was detected by UEAI staining (S1C Fig). StcE treatment was effective as monitored by removal of the MUC1 glycosylated ED from the cellular surface and reduced X409-GFP staining as describe above (Figs 2 and 3). After enzymatic treatment of the Calu-3 cells, a SARS-CoV-2 pseudotyped virus carrying the spike protein and encoding a GFP reporter (SARS2-S pseudotyped VSV-GFP) was added in the absence or presence of an anti-spike monoclonal antibody to confirm spike mediated entry of virus and incubated for 24 hours. StcE treatment enhanced the number of SARS2-S pseudotyped VSV-GFP positive cells, while E447D-treated cells did not show enhanced viral entry. No obvious change in viral infection could be observed after neuraminidase or fucosidase treatment (Fig 3A). In all experimental conditions, pseudoviral infection was completely blocked in the presence of the monoclonal antibody against the SARS-CoV-2 spike protein demonstrating spike-mediated entry in our experimental setup. The GFP signal was quantified using Image J showing a significant 5.4-fold increase in Calu-3 virus infection after StcE treatment and no significant difference after neuraminidase and fucosidase treatment (Fig 3B).

In an independent set of experiments with a luciferase pseudovirus (SARS2-S pseudotyped VSV-Luc), we also observed a 4-fold increase in viral infection after StcE treatment (Fig 3C). As an additional control, we performed the infection with an VSV-G pseudotyped VSV-Luc that lacks the spike protein. We observed enhanced entry of VSV-G pseudotyped VSV-Luc into Calu-3 after StcE and neuraminidase treatment whereas fucosidase treatment had an opposite effect. As expected, the infection could not be blocked with the anti-SARS2-S mAb (Fig 3D). Next, we investigated the effect of mucin removal on infection with the authentic SARS-CoV-2 virus. Calu-3 cells were treated with the enzymes and incubated with SARS-CoV-2 virus in the absence or presence of an anti-spike monoclonal antibody for 8 hours to study initial entry. In line with our pseudovirus experiments, we observed a significant increase in the number of infected cells when cells were treated with StcE mucinase in comparison to control. Again, neuraminidase and fucosidase treatment did not significantly impact viral infection albeit we observed a trend towards increased infection after neuraminidase

## Human Calu-3 respiratory cells

**Fig 3. Removal of the glycosylated MUC1 extracellular domain enhances SARS-CoV-2 entry.** (A) Microscopy images of Calu-3 cells treated with StcE, E447D, neuraminidase or fucosidase infected with SARS2-S pseudotyped VSV-GFP without or with neutralizing monoclonal antibody (mAb) against SARS2-Spike. White scale bars represent 200 μm. (B) Quantification of SARS2-S pseudotyped VSV-GFP signal in Calu-3 cells using EVOS software. StcE treatment resulted in a 5.4-fold increase in infection. (C) Quantification of luciferase signal (RLU) in Calu-3 cells after treatment with indicated enzymes and infection with SARS2-S pseudotyped VSV-Luc in the absence or presence of mAb against spike. A 4-fold increase in RLU value was observed when cells were treated with StcE. (D) Quantification of Calu-3 cell infection with VSV-G pseudotyped VSV-Luc lacking the spike protein. Infection was not blocked by the anti- spike mAb. (E) Infection of Calu-3 cells with authentic SARS-CoV-2 after treatment with indicated enzymes. StcE treatment resulted in a 2-fold increase in infected cell count. Neuraminidase and fucosidase treatment did not significantly impact viral entry. Represented values are the mean ± SEM of three biological replicates performed in triplicate. Statistical analysis was performed by repeated measures one way-ANOVA with Dunnett's post-hoc test. p > 0.05 [ns, not significant], p<0.05 [*], p<0.01 [**], p<0.001 [***], p<0.0001 [****].

treatment (Fig 3E). Together these data demonstrate that removal of glycosylated mucin domains results in increased SARS-CoV-2 infection of lung epithelial cells. No effect on viral entry was observed after removing individual glycans sialic acid and fucose.

Negatively charged molecules such as sialic acid or heparan sulphate (HS) on the cellular surface or extracellular matrix proteoglycans have been described to facilitate viral entry [23,24]. Therefore, we investigated if heparanase treatment to remove HS or neuraminidase treatment to remove sialic acids impacted viral invasion after removal of the MUC1 glycosylated domain with StcE. Calu-3 cells were first treated with StcE, followed by treatment with heparanase or neuraminidase and subsequent viral infection with SARS2-S pseudotyped VSV-Luc. Confocal microscopy confirmed the removal of HS and α-2,6 sialic acid from the surface of Calu-3 cells after heparanase and neuraminidase treatment, respectively (S2A and S2B Fig). Quantification of viral infection showed that the combination treatments did not significantly impact viral invasion compared to StcE only condition (S2C Fig). A small reduction of viral infection was observed when the cells were treated with only heparanase in comparison to the control cells without treatment. All infections in this experiment could be blocked by the mAb demonstrating spike-mediated infection. This result suggests that SARS-CoV-2 entry does not depend on these negatively charged molecules on the cell surface of Calu-3 cells.

## StcE treatment of human primary respiratory cultures enhances SARS-CoV-2 infection

To investigate the role of mucins during SARS-CoV-2 infection in epithelial tissue that more closely resembles the human respiratory surface, we performed infection experiments with human organoid-derived respiratory ALI cultures. First, we determined which mucins were expressed in the ALI cultures. Differentiated cultures were prepared for immunofluorescence staining with and without permeabilization and stained with MUC1, MUC4, MUC5AC and MUC16 antibodies. All mucins were detectable in the permeabilized ALI cultures (Fig 4A). In the non-permeabilized ALI cultures transmembrane mucins MUC1 and MUC16 and traces of MUC5AC networks were detectable suggesting that these mucins are expressed on the cellular surface (Fig 4B). Next, we investigated the efficacy of StcE in cleaving MUC1 and MUC16 from the surface of the airway cultures. Immunoblot analysis of two different donors demonstrated that StcE had effectively cleaved off the glycosylated domain of MUC1 (Fig 4C). The MUC16 antibody recognizes the mucin SEA domain and can therefore not be used to monitor cleavage of the MUC16 glycosylated domain. The X409 domain of StcE was used to detect glycosylated mucin domains and we found that it predominantly colocalized with MUC16 and to a lesser extent with MUC1 (Fig 4D). Colocalization of MUC1 and X406 and MUC16 and X409 was quantified for ALI differentiated cultures of two donors using Mander's overlap coefficient. This analysis demonstrated some colocalization for X409 and MUC1 and a strong colocalization between X409 and MUC16 (Fig 4E). To determine the effect of StcE on surface mucins in general and MUC16 specifically, X409 staining was performed on control ALI cultures and after treatment with StcE and E447D inactive enzyme. A significant reduction of X409 staining could be observed after StcE treatment suggesting cleavage of MUC16 by the mucinase (Fig 4F and 4G).

To determine the effect of mucin removal on SARS-CoV-2 infection, ALI-differentiated airway cultures from two different donors were treated with StcE, E447D or left untreated followed by infection with authentic SARS-CoV-2 virus. StcE treatment led to a significant increase in SARS-CoV-2 infection and replication in both donors, as measured by RNA copies and infectious virus (Fig 5A–5D). Nucleoprotein staining of infected tissues confirmed a high

Human airway organoid-derived ALI cultures

**Fig 4. MUC1 and MUC16 are expressed on the surface of human airway organoid-derived air-liquid interface cultures and decreases upon StcE treatment.** (A) Microscopy of permeabilized human airway organoid-derived air-liquid interface cultures showing combined extracellular and intracellular staining of MUC1, MUC4, MUC5AC and MUC16. (B) Microscopy of live stained air-liquid culture for MUC1, MUC4, MUC5AC and MUC16 without permeabilization. MUC1 and MUC16 are detectable demonstrating expression on the cell surface, whereas MUC4 staining is negative and MUC5AC only stains positive in occasional mucus strands on top of the cells. (C) Immunoblot analysis of MUC1 levels in human airway organoid-derived ALI cultures from donor 1 and donor 2 treated with StcE, E447D or no treatment. The high molecular weight MUC1 is removed upon StcE treatment. (D) Microscopy of MUC1 and MUC16 in permeabilized ALI cultures along with *O*-glycan probe X409-GFP. Arrows indicate co-

localization of X409 with MUC16, but not MUC1. (E) Quantification of colocalization of MUC1 and MUC16 with X409 staining in ALI cultures of two donors as performed in D. Mander's overlap coefficient plots are included in S3 Fig. (F) Microscopy of surface binding of X409 on untreated, 10ug/ml StcE and 10ug/ml E447D treated ALI cultures. All white scale bars indicate 50 μm. (G) Quantification of X409 signal intensity per imaged field from experiment performed in E. Statistical analysis was performed by repeated measures one way-ANOVA with Tukey's post-hoc test. $p < 0.05$ [*].

percentage of virus-infected cells in the StcE condition compared to the control and E447D condition (Fig 5E and 5F).

## Removal of mucin domains enhances spike binding to Calu-3 cells

Next, we investigated if removal of mucin domains directly affected spike and virus attachment to the cellular surface. Calu-3 cells were treated with StcE followed by incubation with purified

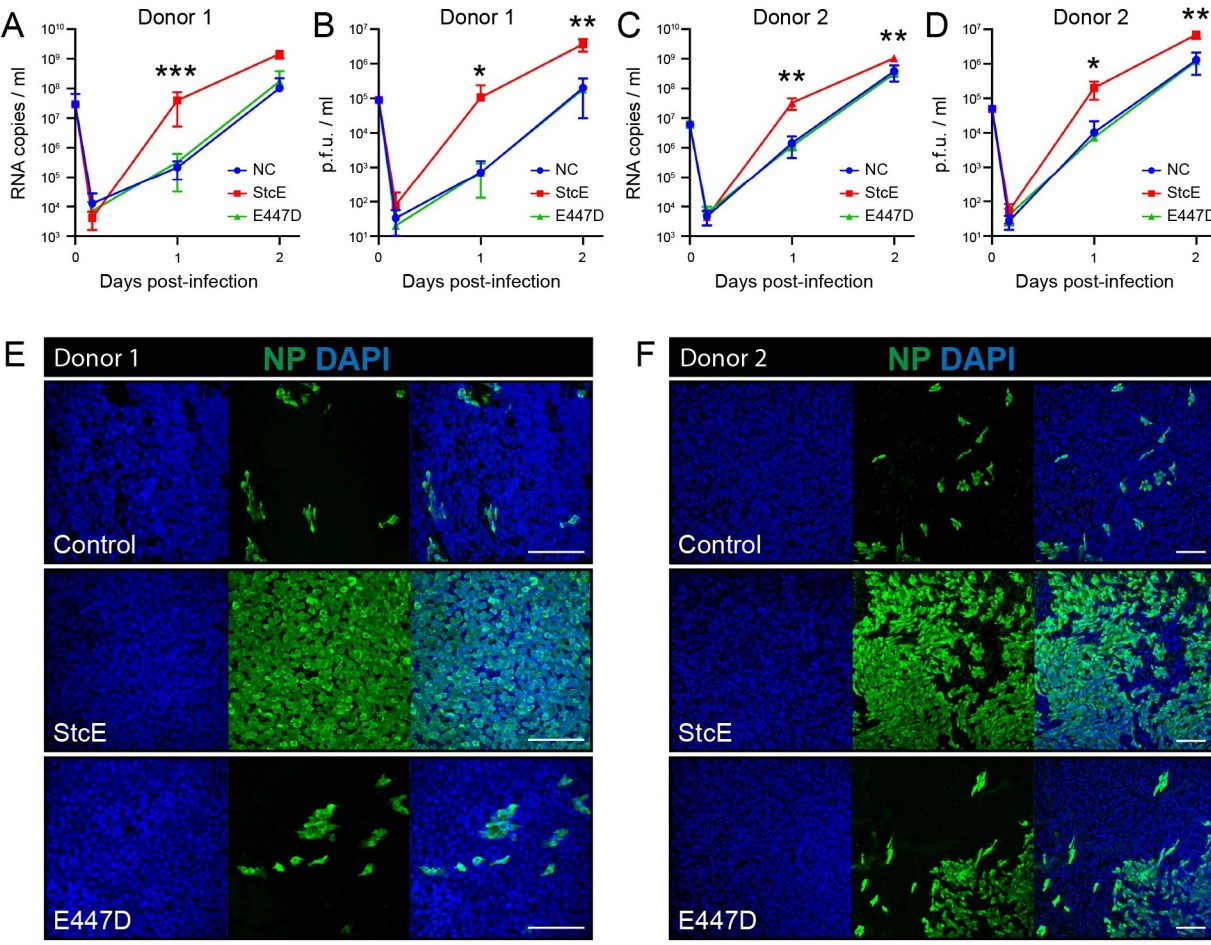

**Fig 5. StcE treatment of human airway organoid-derived air-liquid interface cultures increases SARS-CoV-2 replication.** (A-D) Replication kinetics of SARS-CoV-2 in ALI cultures in terms of RNA copies (A, C) and infectious virus (B, D) in two donors. Represented values are the mean ± SD of three replicates. Statistical analysis was performed for donor 2 by repeated measures two way-ANOVA with Tukey's post-hoc test. $p < 0.05$ [*], $p < 0.01$ [**], $p < 0.001$ [***], $p < 0.0001$ [****]. $p < 0.01$ was found between NC and StcE and E447D and StcE treated cells at 1 day post infection and at 2 days post infection between E447D and StcE treated cells (C). $p < 0.05$ was found between NC and StcE and E447D and StcE treated cells at 2 days post infection (D). (E-F) Microscopy images of untreated, 10ug/ml E447D or 10ug/ml StcE treated cells from donor 1 (E) or donor 2 (F), infected with SARS-CoV-2 at two days post-infection. White scale bars represent 100 μm. NP = nucleoprotein.

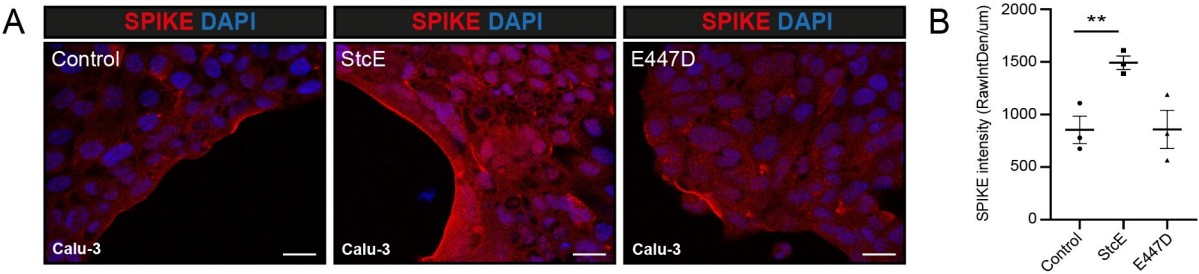

**Fig 6. Removal of the MUC1 extracellular domain increases spike and virus attachment.** (A) Immunofluorescence confocal microscopy of Calu-3 cells incubated with 2.5 ug/ml SARS-CoV-2 spike (Fc-tagged SARS2-S1B-Fc, red) at 4°C for 1 h. Spike was incubated with the cell without permeabilization. Increased spike binding and higher spike signal intensity was observed after treatment with StcE in comparison to E447D treatment and control. White scale bars represent 20 μm (B) Quantification of spike fluorescence signal as depicted in A. Fluorescence intensity along the edge of cell island was determined in control, StcE- and E447D-treated cells using ImageJ. Mean ± SEM raw integrated density/length from three random fields from three independent experiments are plotted. The area of spike binding was significantly higher in StcE-treated cells. White scale bars represent 50 μm.

Fc tagged spike protein (SARS2-S1B-Fc) or SARS2-S pseudotyped VSV-GFP for 1 h at 4°C to monitor attachment and prevent entry. The spike protein was stained without first permeabilizing to prevent intracellular access. In untreated and E447D-treated cells, spike binding was observed in patches along the edge of the cell island while cells treated with StcE showed extensive staining (Fig 6A). Quantification of the fluorescent spike signal on the edges of the cell islands confirmed a significant increase in StcE-treated cells raw spike fluorescence values as determined by integrated density/length (sum of all pixels/μm) (Fig 6B). In a similar experimental setup, we determined the impact of mucin removal on spike binding to ALI cultures of two donors. Large variability in S1B binding was observed among donors with no conclusive effect of StcE treatment. It may be that viral binding in a stratified epithelial cell culture containing a multitude of different cell types (such as our ALI cultures) is more complex than cell lines such as Calu-3 cells and allows additionally binding of S1B independent of ACE2 binding. In addition, differences in the genetic background between donors is likely to affect S1B binding. The increased binding of spike after StcE treatment observed in Calu-3 cells, suggests that in this model mucin extracellular domains form a physical barrier that prevents access to the ACE2 receptor on the epithelial surface. In-depth future studies with organoid cultures of multiple donors are necessary to determine virus-receptor binding and the role of mucins in preventing this interaction in more complex models.

## MUC1 and ACE2 colocalize on the respiratory surface

To understand the spatial relationship between the ACE2 receptor and MUC1, the dominant TM mucin in Calu-3 cells, we performed confocal microscopy on Calu-3 cells stained for ACE2 receptor and the MUC1 extracellular domain with an adjusted protocol that allowed imaging of both proteins. In the monolayer, MUC1- and ACE2-positive cells were observed in distinct scattered distributions and approximately 35% of cells were clearly double positive (Fig 7A). Some cells expressed both proteins at high levels, while other cells expressed high level of either ACE2 or MUC1. MUC1-negative cells most likely express other mucin-like proteins as demonstrated above (Fig 2H). Next, we performed proximity ligation assays (PLA) for ACE2 in combination with one of the MUC1 antibodies that bind to the extracellular domain (214D4 for the glycosylated MUC1-ED and 232A1 for the MUC1-SEA). For both antibody combinations, positive PLA signals could be observed in approximately 40% of cells,

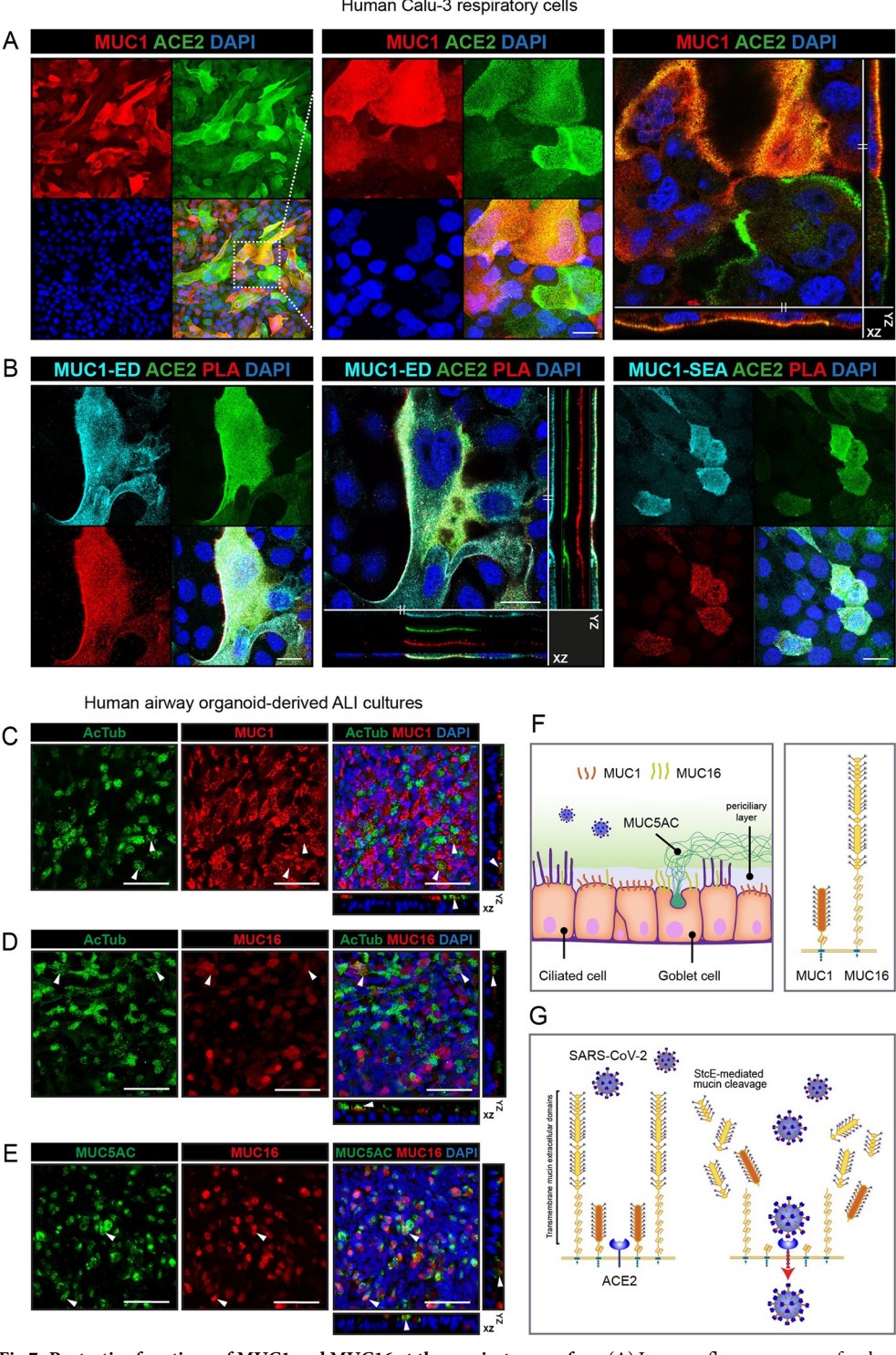

**Fig 7. Protective functions of MUC1 and MUC16 at the respiratory surface.** (A) Immunofluorescence confocal microscopy analysis of expression and localization of ACE2 (green) and TM mucin MUC1 (214D4, red) in Calu-3 cells. (B) Proximity ligation assay (PLA) for MUC1-ED (214D4; light blue) and ACE2 (green) (left) and MUC1-SEA (232A1; light blue) and ACE2 (green) (right) in Calu-3 cells. A positive PLA signal (red) was detectable for both combinations in double positive cells, demonstrating that MUC1 and ACE2 are in close proximity. (C) Immunofluorescence confocal microscopy to determine expression of MUC1 (214D4; red) in ciliated cells (AcTub; green). MUC1 is highly expressed in non-ciliated cells and is expressed in cells with short cilia. (D) Immunofluorescence confocal microscopy for MUC16 (red) and cilia (AcTub; green). MUC16 is expressed in some

non-ciliated cells and ciliated cells. (E) Immunofluorescence confocal microscopy for MUC16 (red) and goblet cells (MUC5AC; green). Several goblet cells are positive for MUC16. White scale bars represent 20 μm for A and B and 50 μm for C-E. (F) Schematic model describing the expression and localization of transmembrane mucins MUC1 and MUC16 in different cell types within the respiratory epithelium. MUC1 is highly expressed in non-ciliated cells and MUC16 is expressed in some ciliated cells and enriched in goblet cells. (G) Schematic model describing the protective functions of the extracellular domains of transmembrane mucins MUC1 and MUC16 during SARS-CoV-2 infection. The extended glycosylated extracellular domains prevent access of the virus to the ACE2 receptor (left). Enzymatic removal of the glycosylated part of the extracellular domains with the StcE mucinase allows the viral spike protein to connect with the ACE2 receptor resulting in viral entry into lung epithelial cells (right).

demonstrating close proximity of <40 nm for MUC1 and ACE2 in a large percentage of respiratory cells (Fig 7B).

The respiratory epithelium consists of non-ciliated cells, ciliated cells and goblet cells that produce soluble MUC5AC. To address specific functions of transmembrane mucins at the respiratory surface, we determined expression of MUC1 and MUC16 in different cell types in air-liquid grown primary respiratory epithelium. MUC1 was highly expressed in the majority of non-ciliated cells and some ciliated cells with short cilia (Fig 7C). MUC16 expression was observed in some ciliated cells (Fig 7D) and in a high percentage of MUC5AC-positive goblet cells (Fig 7E). These results suggest that the transmembrane mucins MUC1 and MUC16 are differentially expressed depending on the cell type and perhaps differentiation state (Fig 7F).

Together our results show that enzymatic removal of mucin extracellular domains including the abundant MUC1 that colocalizes with ACE2 allows more virus attachment to cells and thus increases infectivity. Different cell types most likely have their own specific mucin glycocalyx. We propose that the glycosylated domains of transmembrane mucins such as MUC1 and MUC16 form a barrier that prevents SARS-CoV-2 invasion at the respiratory surface (Fig 7F and 7G).

## Discussion

The mucosal barrier is the body's first line of defense and offers protection from infection by pathogens. Mucin proteins are known for their barrier properties but can also serve as attachment sites for bacterial and viral pathogens. The findings presented in this study indicate that extracellular mucin domains play a substantial protective role during SARS-CoV-2 infection at the respiratory surface. ACE2-positive cells in the respiratory epithelium express different combinations of mucin genes including MUC1, MUC4, MUC16, MUC5AC and MUC5B of which MUC1 is the most abundantly expressed mucin across different cell types. In this study, we demonstrate that human lung epithelial Calu-3 cells expressed high levels of MUC1, while TM mucins MUC4 and MUC16 and secreted mucin MUC5AC were barely detectable (Figs 1 and 2). In organoid-derived airway cultures, MUC1 and MUC16 are expressed on the surface and some secreted MUC5AC is detectable (Fig 4). In different SARS-CoV-2 infection studies with both cell models, we demonstrate that enzymatic removal of extracellular mucin domains, but not individual sialic acid or fucose sugars, enhances viral infection (Figs 3 and 5). Removal of glycosylated mucin domains from the cellular surface increased binding of purified spike protein to the cellular surface (Fig 6). We demonstrate for the first time that MUC1 and ACE2 colocalize on the apical surface of Calu-3 cells and that MUC1 and MUC16 have unique expression patterns in different cell types in our air-liquid differentiated airway cultures (Fig 7). We propose a model in which glycosylated extracellular mucin domains form a protective layer above the underlying ACE2 receptor thereby preventing access of the virus to the receptor (Fig 7).

There is growing evidence that both soluble and transmembrane mucins play important roles during SARS-CoV-2 infection, but whether their contributions are protective or are facilitating pathogenicity at different stages of disease is still under debate. Mucus hypersecretion and accumulation most likely have a negative impact on disease development due to reduced MCC and mucus plugging [25], but studies into the roles of transmembrane mucins point in different directions. In the course of COVID-19, elevated levels of gel-forming MUC5AC and shed MUC1 can be detected in sputum aspirated from the trachea of patients [26] and high production of MUC5AC was observed in SARS-CoV-2 infected primary respiratory cultures [27]. The MUC5B genetic variant rs35705950 is associated with higher expression of the soluble mucin MUC5B and underrepresented in COVID-19 patients compared to healthy individuals, suggesting a protective role for MUC5B [28]. In aged individuals, decreased mucus production and weakened MCC might contribute to the higher susceptibility of SARS-CoV-2 [29]. Our findings on a protective role for endogenously expressed TM mucins MUC1 and MUC16 during SARS-CoV-2 infection are in line with a recent genome-scale CRISPR loss- and gain-of-function (GOF) study for SARS-CoV-2 entry in human lung epithelial cells overexpressing TM mucins [30]. Overexpression of TM mucins MUC1, MUC4 or MUC21 reduced infection by SARS-CoV-2 compared to cells with a non-targeting guide (NTG). The study also demonstrated that enzymatic removal of overexpressed MUC4 resulted in increased viral entry. An important role for MUC4 was also observed during SARS-CoV-1 infection *in vivo* where female MUC4 knockout mice that had enhanced inflammatory cytokine responses and poor prognosis compared to wild type mice [31]. These different studies and our own results imply that different TM mucins might have similar protective functions in different cell types during SARS-CoV-2 infection. The challenge in the field is now to understand which TM mucins are expressed where and how this localization relates to ACE2-mediated entry of the virus. It was previously reported that ACE2 is mostly expressed on the apical basal surface of ciliated cells [32] but another study found ACE2 expression on the cilial shafts [33]. A recent study also found ACE2 expression on cilia and demonstrated that SARS-CoV-2 virions attach to motile cilia before cellular entry [34]. Interestingly, the omicron variant has higher affinity for motile cilia [34]. While ciliated cell may be a main target of SARS-CoV-2 virions, goblet cells can also be infected [27]. Our experiments show heterogeneous expression of MUC1 and ACE2 in Calu-3 cells and point at a unique expression pattern for MUC1 in non-ciliated cells and cells with short cilia, while MUC16 was somewhat found in ciliated cells and predominantly expressed in goblet cells. These results are largely in line with an extensive study into TM mucin expression and localization in human tracheobronchial epithelial cultures were MUC1 was found to be highly expressed on microvilli and on basal epithelial cells [6]. In this study, MUC4 localized on the cilia and MUC16 was highly expressed on goblet cells. Future research should address the relationship between TM mucins and different surface receptors relating to viral and bacterial infection.

In addition to the protective functions of mucins during SARS-CoV-2 initial infection, there is also emerging evidence that overexpression of different mucins is correlated with severe disease. A MUC1 gene variant that leads to increased expression was one of the few significant loci associated with severe COVID in a large-scale GWAS study. The functional consequences of this gene variation need to be addressed, but the authors suggest that mucins could have a clinically important role in the development of critical illness in COVID-19 [35]. This was in line with another study that found increased MUC1 mRNA to be associated with critical disease [36]. Single cell sequencing data of COVID-19 patients demonstrated that transmembrane mucins MUC1, MUC4, MUC13 and MUC21 are all highly upregulated in patients with active disease [30] and also in blood samples MUC1 and MUC2 mRNA expression was significantly elevated in critical and mild COVID-19 while MUC16, MUC20 and

MUC21 were significantly downregulated in severe COVID [37]. Compound R406, the active metabolite of FDA-approved Fostamatinib that inhibits MUC1 expression is now in clinical trials for hospitalized patients with advanced COVID-19 [38]. At this point, we lack the critical insight to conclude if transmembrane mucins in general or MUC1, MUC4 or MUC16 specifically are protective or contributing to disease severity during different stages of pathogenesis within the complexity of the body. It is evident that establishing the function of specific mucins during *in vivo* infection is an important future challenge.

The studies that are currently available underscore the importance of extracellular domain of transmembrane mucins during viral entry. Our confocal microscopy analysis indicates that ACE2-positive cells in the respiratory epithelium often, but not always, express MUC1 and that both proteins colocalize on the apical surface. Enzymatic removal of the MUC1 glycosylated domain did not affect the underlying SEA domain or cytoplasmic tail and ACE2 expression remained detectable. As was previously hypothesized, it is possible that MUC1 and ACE2 interact and/or are in the same protein complex on the respiratory surface [39]. Our Calu-3 data indicate that steric hindrance by glycosylated extracellular mucin domains prevents the virus from reaching the ACE2 receptor (Fig 6). This is in line with a recent study that used mucin mimetics glycopolymers that were capable of shielding surface receptors [40]. In a previous study, we have shown that MUC1 ED alters the cell membrane of non-polarized epithelial cells to tubulated morphology and reduce β1-integrin-mediated bacterial invasion (19). In the present study, we have not observed any influence of MUC1 ED on membrane architecture in Calu-3 cells or organoid-derived airway cultures.

Different studies describe that for viral entry SARS-CoV-2 benefits from negatively charged residues like sialic acid-containing glycans or membrane glycosaminoglycans such as heparan sulfate proteoglycans on the cell surface [41–43]. In contrast with this findings, another study reported that neuraminidase treatment of Calu-3 cells only modestly increased SARS-CoV-2 infection [44]. In our live virus experiments we did not observe a significant increase in SARS-CoV-2 infection after neuraminidase or fucosidase treatment. We addressed if the negatively charged sialic acids or heparan sulfates were important for viral entry after removal of the glycosylated mucin domain. Consecutive treatment with StcE and neuraminidase or heparinase was performed but did not result in a difference in viral entry (S2 Fig). Differences in viral dependence on negatively charged surface molecules maybe be explained by levels of ACE2 and TMPRSS2 protease expression and accessibility of the receptor for the viral spike protein in different cell systems. Therefore, our findings reveal that during infection of human respiratory Calu-3 cells the MUC1 extracellular domain rather than individual mucin glycans prevents the binding of SARS-CoV-2 to the underlying receptor.

Overproduction and excess accumulation of gel-forming mucins in the lungs of COVID-19 patients can lead to airway obstruction and eventually cause life-threatening acute respiratory distress syndrome [45–47]. Several studies are focusing on the reduction of mucin expression overall as a therapeutic strategy [38,48]. In future studies, the role of transmembrane mucins MUC1, MUC4 and MUC16 in disease development should be taken into consideration. A tailored approach that boosts expression of protective transmembrane mucins but reduces secretion of soluble mucins could be an attractive future strategy to prevent infection with SARS-CoV-2 or other respiratory pathogens and improve disease outcome.

## Methods

### Ethics statement

Adult human lung tissue was obtained from non-tumor lung tissue obtained from patients undergoing lung resection. Lung tissue was obtained from residual, tumor-free, material

obtained at lung resection surgery for lung cancer. The Medical Ethical Committee of the Erasmus Medical Center Rotterdam granted permission for this study (METC 2012–512) and formal written consent was obtained was obtained from the patients/donors.

## Single cell analysis

Normalized counts and metadata from previously published single cell RNA-sequencing data of healthy human airway epithelium [16] were downloaded from https://www.genomique.eu/cellbrowser/HCA/. Dimensionality reduction was done using the Seurat Package [49] in Rstudio (version 1.2.5019), starting with a principle component analysis. After visual inspection of the principal components using and elbow plot, the first twenty components were used for graph-based clustering analysis. Clusters of cells were then visualized as diffusion maps (uMAPs). To determine gene expression in ACE2- and TMPRSS2-positive versus negative cells we created two additional metadata slots, in which normalized transcript counts of these genes above 0 were considered positive. Then, cell type assignment and normalized expression of a panel of genes of interest was determined by sub-setting single or double-positive epithelial cells.

## Cell culture

Calu-3 cells (ATCC Catalog # HTB-55), HEK-293T (ATCC Catalog # CRL-3216) and BHK-21 cells (ATCC Catalog # CCL-10) cells were routinely grown in 25 cm$^2$ flasks in Dulbecco's modified Eagle's medium (DMEM) containing 10% fetal calf serum (FCS) at 37˚C in 5% $CO_2$.

## Human airway organoid culture and differentiation

Human bronchiole and bronchus stem cells were isolated and maintained as described previously [50,51], using a protocol adapted from Sachs and colleagues [52]. Organoids were dissociated using TrypLE express (Gibco) into single cells and plated on Transwell membranes (StemCell) coated with rat tail collagen type I (Fisher Scientific) in Pneumacult-ALI medium (StemCell) and airway organoid medium at a 1:1 ratio as described before [50,51]. Upon confluency, cells were differentiated at an air-liquid interface in 100% Pneumacult-ALI medium for 3–6 weeks. Medium was replaced every 5 days.

## Production of SARS-CoV-2 pseudotyped virus and virus neutralization assay

For this study, we used SARS-CoV-2 pseudovirus with spike sequences of the original pandemic virus. The pseudotyped vesicular stomatitis virus (VSV) was produced by using the protocol of Whitt [53]. The detailed protocol of the production of pseudotyped VSV, SARS2-Spike pseudotyped VSV virus and virus neutralization assay is described in S1 Methods. The optimal working concentration of SARS2-Spike pseudotyped VSV particles (SARS2-S pseudotyped VSV-GFP and SARS2-S pseudotyped VSV-Luc) was determined by viral titration assay on Calu-3 cells.

## Production of authentic SARS-CoV-2 virus stock

SARS-CoV-2 (isolate BetaCoV/Munich/BavPat1/2020; European Virus Archive Global #026V-03883; kindly provided by Dr. C. Drosten) was propagated on Calu-3 cells in Opti-MEM I (1X) + GlutaMAX (Gibco), supplemented with penicillin (100 IU/mL) and streptomycin (100 IU/mL) at 37˚C in a humidified $CO_2$ incubator. Stocks were produced as described previously [54]. A detailed description of virus production can be found in S1 Methods.

## Enzyme treatment of Calu-3 cells

StcE and StcE-E447D were expressed and purified as described previously [20]. For mucinase treatment, Calu-3 cells were treated with 2.5 ug/ml of StcE or its inactive mutant E447D in 10% FCS media for 3 h at 37˚C and washed with DPBS. Desialylation of Calu-3 cells was achieved by incubating cells grown in a 96 well plate or 24-well plate or 6 well plate with 100 U/mL α2–3,6,8,9 neuraminidase A (P0722L, NEB) in 10% FCS media for 3 h at 37˚C. For fucosidase treatment of Calu-3 cells, 0.4 U/ml of α-(1–2,3,4,6)-L-Fucosidase (E-FUCHS; Megazyme) was added to the cells and incubated for 3 h at 37˚C. To remove heparan sulfate (HS), 0.1 U/ml heparinase III (H8891-5UN, Sigma) was applied as described for the other enzymes. After enzyme treatment, cells were washed thrice with DPBS and used for subsequent experiments.

## SARS-CoV-2 infection assays on Calu-3 cells

For infection experiments, Calu-3 cells were grown in 96-well plates and allowed to reach around 90% confluency. Then, cells were treated with enzymes for 3 h at 37˚C and 5% $CO_2$, before they were inoculated with SARS2-S pseudotyped VSV-Luc or SARS2-S pseudotyped VSV-GFP. At 20 h post-infection, culture supernatants were aspirated, washed with DPBS, and cells were lysed by overnight incubation with *Renilla* luciferase assay lysis buffer (Promega) at -80˚C. The next day, cell lysates were thawed, thoroughly resuspended, and transferred to white, opaque-walled 96-well plates and relative luminescence unit (RLU) was measured. *Renilla* luciferase activity was determined using the Luciferase Assay Systems (Promega) according to the manufacturer's instructions. Raw luminescence values were recorded as counts per 5 seconds by Berthold Centro LB 942 plate luminometer. For SARS2-S pseudotyped VSV-GFP mediated infection, GFP positive signal captured using an EVOS microscope (Thermo Scientific) at 4X magnification and quantified using EVOS software. For infection experiments with the authentic SARS-CoV-2 virus, Calu-3 cells were prepared as described above and inoculated with approximately 200 pfu of SARS-CoV-2. At 8 h post-infection, cells were washed in PBS, fixed in formalin, permeabilized in 70% ethanol and washed in PBS again. Immunofluorescent stainings were performed as described for SARS-CoV-2 stock production and scanned plates were analyzed using ImageQuant TL software. All work with infectious SARS-CoV-2 was performed in a Class II Biosafety Cabinet under BSL-3 conditions at Erasmus Medical Center.

## SARS-CoV-2 infection assays on human airway organoid-derived ALI culture

Prior to infection, ALI cells were washed three times with Advanced DMEM/F12 (Gibco) supplemented with Hepes (20mM, Lonza), Glutamax (Gibco) and Primocin (200ug/ml; Invivogen) (AdDF+++). Cells were pretreated for 3 hours with either 10ug/ml E447D or StcE in AdDF +++ or AdDF +++ alone. Cells were washed three times with AdDF +++ and infected with an MOI of 0.01 of SARS-CoV-2 for 4 hours, at which time cells were washed three times with ADdF +++ and remained on ALI for the duration of the experiment. Apical washes were collected at 4, 24 and 48 hours post infection and viral loads were detected in different treatment conditions. After the last collection cells were fixed in 4% formalin for 20 minutes, followed by 70% ethanol for 20 minutes. Plates were exported from the BSL-3 in ethanol for subsequent staining.

### Human airway organoid-derived ALI infection growth curves

All samples were thawed and centrifuged at 2000x g for 5 min to spin down mucus and cellular debris. Supernatant was used for subsequent analysis. Virus titrations to determine pfu/ml were performed as described in S1 Methods for determining SARS-CoV-2 titres. RNA extraction was performed by adding 60 μl of sample to 90 μl MagnaPure LC Lysis buffer (Roche) for 10 minutes. Fifty μl Agencourt AMPure XP beads (Beckman Coulter) were added and incubated for followed by two washes on a DynaMag-96 magnet (Invitrogen) and elution in 30 μl ultrapure water. All steps were performed at room temperature. RNA copies were determined by qRT-PCR using primers targeting the E gene (51) and comparison to a standard curve.

### Confocal microscopy

Cells were grown on coverslips up to 80% confluency were analyzed by immunofluorescent staining. Cells were washed twice with DPBS and fixed with 4% paraformaldehyde in PBS (Affymetrix) for 20 min at room temperature and fixation was stopped with 50 mM $NH_4Cl$ in PBS for 10 min. The staining procedure and antibody details are described in S1 Methods.

### Human airway organoid-derived ALI fluorescent staining

ALI inserts infected with SARS-CoV-2 were fixed in 4% formalin for 20 minutes followed by 70% ethanol for 20 minutes and washed in PBS. Uninfected ALI inserts were either fixed and permeabilized with 0.1% triton-X in 10% normal goat serum (NGS) in PBS or stained live on ice. All inserts were blocked in 10% NGS in PBS for an hour followed by primary antibody incubation overnight or for 4 hours on live cells on ice: rabbit anti-SARS-CoV-2 nucleoprotein (Sinobiological, 40143-T62, 1:1000), mouse anti-acetylated tubulin (1:100, Santa Cruz), mouse anti-MUC1 ED, mouse anti-MUC4, mouse anti-MUC5AC or mouse anti-MUC16 (source and dilution was mentioned in S1 Methods). After incubation with primary antibody, live cells were fixed and washed with PBS. All other inserts were washed with PBS. For secondary antibody incubation, Alexa Fluor 568-conjugated goat α-mouse IgG, Alexa Fluor 488-conjugated goat α-rabbit IgG, Alexa Fluor 647-conjugated goat α-mouse IgG or X409-GFP (source and dilution mentioned in S1 Methods) was used. Secondary antibodies were incubated for one hour. All antibodies were diluted in 10% NGS in PBS. After secondary antibody incubation cells were washed with PBS and stained for nuclei using DAPI diluted in PBS. After 30 minutes incubation cells were washed in PBS and mounted in Prolong Antifade (Invitrogen) mounting medium. ALI culture confocal microscopy was performed on an LSM700 confocal microscope using ZEN software (Zeiss). Representative images are maximum intensity projections taken from Z-stacks. Mean signal intensities and Mander's overlap coefficients of MUC1, MUC16 and X409 were analysed using ZEN software.

### Western blotting

Calu-3 cells were grown in 6-well plates for 7 days before enzyme treatment. Enzyme-treated cells were washed thrice with cold DPBS and collected with a scraper. The cell suspension was centrifuged at 5,000 rpm for 5 min at 4°C. Cell pellets were resuspended with 100 μl 1% SDS in presence of a Halt protease inhibitor cocktail and 0.5 M EDTA solution (Thermo Fisher) and cells lysed mechanically by scratching. Protein concentrations were measured using a BCA protein assay kit (23235#, Pierce Company). For detection of the MUC1 ED, 5% mucin gels and a boric acid-Tris system were used as described previously [55]. For dotblot analysis, protein lysates were directly spotted on nitrocellulose membrane, dried and blocked and incubated with antibodies as for regular western blots. α-MUC1-ED antibody 214D4 was used to

detect MUC1 at a dilution of 1:1,000 in TSMT buffer. For detection of the CT of MUC1, 12% SDS-PAGE gel and α-MUC1-CT antibody CT2 was used. For ACE2 detection, 10% SDS-PAGE gel and anti-ACE2 antibody (1:1,000, HPA000288, Sigma-Aldrich) was used. Actin was detected using α-actin antibody (1:5,000; bs-0061R, Bioss). Secondary antibodies used were α-mouse IgG secondary antibody (1:10,000; A2304, Sigma), α-Armenian hamster IgG (1:10,000; GTX25745, Genetex) and α-rabbit IgG (1:10,000; A4914, Sigma). Blots were developed with the Clarity Western ECL kit (Bio-Rad) and imaged in a Gel-Doc system (Bio-Rad).

## Statistical analysis

For all experiments, at least three independent biological replicates were performed. Values are expressed as the mean ± SEM of three independent experiments performed in triplicate. Repeated measures one way-ANOVA with Dunnett's or Tukey's or two way-ANOVA with Tukey's post-hoc was applied to test for statistical significance. P values of 0.05 or lower were considered statistically significant. Symbols used are $p > 0.05$ (ns, not significant), $p < 0.05$ (*), $p < 0.01$ (**), $p < 0.001$ (***), $p < 0.0001$ (****). The GraphPad Prism 9 software package was used for all statistical analyses.

## Supporting information

**S1 Fig. Characterization of expression of mucins and mucin glycans on Calu-3 cells.** (A) Immunofluorescence confocal microscopy of 4-days grown Calu-3 cells revealed very limited expression of MUC16 (α-MUC16 ED, green). (B) Immunofluorescence confocal microscopy images for α-2,6 sialic acid (SNA, green) and α-2,3 sialic acid (MALII, green) levels after neuraminidase treatment and fucose (UEAI, green) after fucosidase treatment of Calu-3 cells. Nuclei were stained with DAPI (blue). White scale bars represent 20 μm.
(TIF)

**S2 Fig. Heparinase or neuraminidase treatment in combination with mucinase does not affect SARS-CoV-2 entry.** (A) Confocal microscopy image showing levels of heparin sulfate (F69-3G10, green) in control, heparinase-treated and StcE/heparinase-treated Calu-3 cells. (B) Confocal microscopy image showing levels of α-2,6 sialic acid (SNA, green) control, neuraminidase-treated and StcE/neuraminidase-treated Calu-3 cells. Nuclei were stained with DAPI (blue). White scale bars represent 20 μm. (C) Luciferase quantification of viral infection of Calu-3 cells were treated with StcE for 3 h at 37°C followed by heparinase or neuraminidase for an additional 3 h at 37°C and infection with VSVΔG-Rluc*SARS2-Spike without or with monoclonal antibody (mAb) against SARS2-Spike. No significant changes in RLU values were observed in any of the cases. Represented values are the mean ± SEM of three biological replicates performed in triplicate. Statistical analysis was performed by repeated measures one way-ANOVA with Dunnett's post-hoc test. $p > 0.05$ [ns, not significant], $p < 0.05$ [*], $p < 0.01$ [**], $p < 0.001$ [***], $p < 0.0001$ [****].
(TIF)

**S3 Fig. Quantification of MUC1 and MUC16 colocalization with X409 probe in airway ALI cultures.** Organoid-derived airway ALI cultures were stained for MUC1 or MUC16 in combination with X409 as depicted in Fig 4D. Colocalization of MUC1 and MUC16 with X409 was quantified cultures from dono1 2 (A) and donor 3 (B) by plotting Mander's overlap coefficient using ZEN software. The summary of this quantification is depicted in Fig 4E.
(TIF)

**S1 Methods. Describing supplementary methods on production of pseudotyped vesicular stomatitis virus (VSV) and SARS2-Spike pseudotyped VSV virus, production of authentic SARS-CoV-2 virus stock, PLA assay for MUC1 and ACE2 and detailed confocal microscopy methods.**
(DOCX)

## Acknowledgments

We thank Wentao Li and Arno van Vliet for assistance with virus production. We thank Robert P. de Vries (Faculty of Pharmaceutical Sciences, Utrecht University) for providing expert advice and reagents for the heparan sulfate experiments. We thank Robbert Rottier for providing human adult lung material. The X409-GFP reagent was kindly provided by Yoshiki Narimatsu (Copenhagen Center for Glycomics). We thank Celia Segui-Perez for expert advice on PLA assays, Dave Lifka for assistance with microscopy quantification and Daphne Stapels for valuable input.

## Author Contributions

**Conceptualization:** Maitrayee Chatterjee, Bart L. Haagmans, Karin Strijbis.

**Data curation:** Maitrayee Chatterjee, Anna Z. Mykytyn.

**Formal analysis:** Maitrayee Chatterjee, Liane Z. X. Huang, Anna Z. Mykytyn, Bart Westendorp.

**Funding acquisition:** Bart L. Haagmans, Karin Strijbis.

**Investigation:** Maitrayee Chatterjee, Liane Z. X. Huang, Anna Z. Mykytyn, Chunyan Wang, Mart M. Lamers, Bart Westendorp, Richard W. Wubbolts.

**Methodology:** Maitrayee Chatterjee, Liane Z. X. Huang, Anna Z. Mykytyn, Chunyan Wang, Mart M. Lamers, Bart Westendorp, Richard W. Wubbolts, Berend-Jan Bosch, Karin Strijbis.

**Project administration:** Maitrayee Chatterjee, Liane Z. X. Huang, Anna Z. Mykytyn.

**Resources:** Chunyan Wang, Mart M. Lamers, Jos P. M. van Putten, Berend-Jan Bosch, Bart L. Haagmans.

**Supervision:** Jos P. M. van Putten, Berend-Jan Bosch, Bart L. Haagmans, Karin Strijbis.

**Validation:** Maitrayee Chatterjee, Liane Z. X. Huang, Anna Z. Mykytyn.

**Visualization:** Maitrayee Chatterjee, Liane Z. X. Huang, Anna Z. Mykytyn, Bart Westendorp, Richard W. Wubbolts, Karin Strijbis.

**Writing – original draft:** Maitrayee Chatterjee.

**Writing – review & editing:** Anna Z. Mykytyn, Jos P. M. van Putten, Bart L. Haagmans, Karin Strijbis.

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
