## [Decision Letter · Decision Letter 0]

13 Mar 2023

Dear Dr. Strijbis,

Thank you very much for submitting your manuscript "Glycosylated extracellular mucin domains protect against SARS-CoV-2 infection at the respiratory surface" for consideration at PLOS Pathogens. As with all papers reviewed by the journal, your manuscript was reviewed by members of the editorial board and by several independent reviewers. In light of the reviews (below this email), we would like to invite the resubmission of a significantly-revised version that takes into account the reviewers' comments.

Thank you for the manuscript. Three colleagues have commented with specific points to consider for a resubmission. One thing that should be emphasized in a revision is how this work differentiates itself from other already published manuscripts on mucins and SARS-CoV-2, as pointed out by several reviewers. Please let me know if there are any questions or concerns.

We cannot make any decision about publication until we have seen the revised manuscript and your response to the reviewers' comments. Your revised manuscript is also likely to be sent to reviewers for further evaluation.

Sincerely,

Matthew B. Frieman

Guest Editor

PLOS Pathogens

Sara Cherry

Section Editor

PLOS Pathogens

Kasturi Haldar

Editor-in-Chief

PLOS Pathogens

orcid.org/0000-0001-5065-158X

Michael Malim

Editor-in-Chief

PLOS Pathogens

orcid.org/0000-0002-7699-2064

Thank you for the manuscript. Three colleagues have commented with specific points to consider for a resubmission. One thing that should be emphasized in a revision is how this work differentiates itself from other already published manuscripts on mucins and SARS-CoV-2, as pointed out by several reviewers. Please let me know if there are any questions or concerns.

Reviewer's Responses to Questions

**Part I - Summary**

Reviewer #1: Understanding host-pathogen interactions at the cellular surface are essential for improving our knowledge surrounding infection processes and host-defences. This expert team asks the important question as to whether the cell-surface and gel-forming mucins of the respiratory mucosa aid or hinder the ability for SARS-CoV-2 to infect cells. Using appropriate mucinases, fucosinases, neuroaminidases, and antibody controls the authors conclusively show that the cell-surface MUC1 extracellular domain provided a barrier-type role and aided in limiting SARS-CoV-2 infection of the underlying cell. When MUC1 was expressed in it’s natural form, less SARS-CoV-2 was able to gain access to ACE2 (the virus receptor), while removal of the extracellular domain of MUC1 enhanced ACE2 access and infection. The authors also correlate their findings with human ALI-organoid cultures, proving biological relevance. The single-cell RNA seq data visualized in diffusion maps matched with the experimental data, where the expected expression of each mucin subtype was shown to be distributed as expected in the ALI and Calu3 cell types. The discussion places the findings of the paper in relevant context – where care is taken to explain that while MUC1 may provide a novel therapeutic target to limit SARS-CoV-2 infection, treatments that alter the homeostasis of this glycoprotein during the course of infection may alter other important functions that MUC1 plays in immune regulatory responses to the infection, but clearly this cs-mucin plays an important role in COVID severity.

This is a high quality manuscript which clearly answers an important and currently high impact research question.

Reviewer #2: In this manuscript, Chatterjee et al. investigate the role of tethered mucins and, specifically, their terminal glycans during SARS-CoV-2 infection. They utilize both Calu3 monolayer cultures and primary airway epithelial organoids (air-liquid interface cultures) along with various enzymatic treatments to remove glycosylated extracellular mucin domains, specific sialic acids, or fucose molecules to determine whether these mucins/glycans may promote adherence, or act as a barrier to infection. In conclusion, the authors propose a model in which mucins block access to ACE2 and thereby restrict infection. While the paper is well-written, the majority of the paper is spent characterizing mucin expression after StcE treatment, which remains somewhat unclear due in part to the limited availability of antibodies that detect specific domains of specific mucins. Further, it is unclear why the authors have chosen to focus much of the paper on Calu3 cells which lack cilia and fail to express all the major tethered airway mucins (e.g. MUC16). The most exciting result is found in Figure 5, where StcE treatment of airway cultures yields a significant increase in infection. However, the paper does not resolve which tethered mucin(s) may be responsible for the barrier function. This, in conjunction with the fact that several aspects of the work have been investigated before (e.g. ACE2 and mucin expression in tissue and airway models, impact of NA treatment on infection, etc.), significantly limits the impact and novelty of the study.

Reviewer #3: This submission reports findings that transmembrane mucins restrict SARS-CoV-2 entry. The report has convincing data that demonstrates mucin interference with virus-cell binding. Mucins have been investigated in many laboratories for their effects on virus infections. The submission also provides findings on the roles of sialic acid, fucose and heparin as SARS-CoV-2 susceptibility factors. While these latter findings were negative (enzymatic removal had no effect on infection), they may be significant in addressing the credibility of previous reports claiming that these carbohydrates are involved in SARS-CoV-2 entry.

**Part II – Major Issues: Key Experiments Required for Acceptance**

Reviewer #1: There are no major issues for this manuscript that I can tell.

Reviewer #2: • The authors note in the introduction that MUC16 is associated with cilia (citing a review from Hattrup and Gendler published in 2008). However, more recent work by Kesimer et al. (doi: 10.1038/mi.2012.81) has suggested MUC16 localizes to goblet cells.

• In Figure S1 and 1, the authors detect little MUC16 in Calu3 cells and fail to detect MUC4 on the surface of Calu3 cells. They suggest that MUC4 has intracellular localization; however, it may also be that the antibody used does not detect the mature glycosylated form of MUC4. Can the authors confirm whether or not this is the case? Further, if the data are in fact correct and there is no MUC16 or MUC4 expression on the surface of Calu3 cells, then these cells do not seem to be a good model system in which to interrogate the impact of tethered mucins on SARS-CoV-2 infection.

• At line 138 the authors note a partial loss of MUC1-ED detection after NA treatment, but indicate MUC1 was still detectable on the surface of cells by IF and thus, the western results are likely due to a change in epitope recognition. While not a focus of the manuscript, the IF data here appear to be missing. Also, given the partial loss, some MUC1 expression on the cell surface would be expected, and since the IF data does not appear to have been quantified in any way, the rationale for interpreting the western blot in this way is unclear.

• The results in Fig 2H are a bit confusing – the loss of MUC1 detection is obvious after StcE treatment, but X409 binding does not appear altered. Shouldn’t X409 binding be lost with StcE treatment? The authors note that StcE targets an O-glycosylated serine-threonine motif that is virtually absent in non-mucin proteins (line 123). If this is true, then what is X409 binding to in Fig 2H on the surface of Calu3 beyond MUC1? On a related note, please separate the channels in the confocal microscopy (Figure 2H and 2I) as some of the signal is difficult to see in the merge.

• Similarly, in Fig 4D,E, why doesn’t X409 staining co-localize with MUC1 as it seems to do in Fig 2H? The colocalization with MUC16 in Fig 4E is also not convincing. It seems that both MUC1 and MUC16 should react with X409.

• It is unclear why the authors return to Calu3 cells in Figure 6. Why not incubate the airway cultures with Fc-tagged spike? A similar experiment was recently successfully performed using quantum dot-conjugated SARS-CoV-2 RBD in airway cultures to assess binding (doi: 10.1016/j.cell.2022.11.030).

• In Fig 6E, the authors conclude that spike-positive pseudoviral particles preferentially bound to MUC1-negative regions. This conclusion appears to be based on one single region of spike staining, and is thus not well supported.

• The authors conclude that mucins block access to ACE2 and thereby restrict infection, and the current data in Calu3 cells would suggest this block is primarily due to MUC1. How do the authors fit this model with their data showing that MUC1 and ACE2 only partially co-localize on the apical surface of Calu3 and with published work in human airway cultures placing ACE2 on the apical membrane of ciliated cells (https://doi.org/10.1128/JVI.79.24.15511-15524.2005) or on cilial shafts (DOI: 10.1038/s41467-020-19145-6) while MUC1 is found on microvilli, predominantly on non-ciliated cells?

• Overall, the manuscript would be strengthened by including image quantification or colocalization analysis, as well as additional means of mucin removal (e.g. knockout of individual or all tethered mucins).

Reviewer #3: 1. One concern with this submission is that many of the findings have been reported previously, particularly in PMID: 35879412 (ref 29), which communicates experiments and results strikingly similar to those in this submission. While these prior works were discussed on lines 280-300, further improvements in the submission would come with clear indication of the findings that are new, with clearer statements of their significance in advancing current understanding of mucins in SARS-CoV-2 infection.

2. Interpretation of findings and the Fig 6F model appeal to the close proximity of MUC1 with ACE2. While there are data showing MUC1 – ACE2 coexpression, the discussion and model in Fig 6F could be further supported with higher resolution imaging. Might proximity ligation assays generate more convincing evidence of MUC-ACE2 colocalization?

3. S1B-Fc is far smaller than SARS-CoV-2 particles. Is S1B-Fc a suitable probe for TM MUC-mediated steric interference with ACE2 binding, given that it is small and possibly more penetrant into the mucus layer than virions? (Also, a minor question about the SARS-pseudotyped VSV that was used to probe ACE2 binding – is this a VSV that has GFP incorporated into the virus particles, or is it a VSV that expresses GFP after S-mediated transduction? Probably it is the former, but perhaps it should be overtly specified as the virus is used as a probe for ACE2 binding.)

4. While the discussion section is interesting and informative, there could be mention and discussion of PMID: 36580912, which shows restricted SARS-CoV-2 cell binding being overcome by attachment at the tips of cilia that extend beyond the mucus layers.

**Part III – Minor Issues: Editorial and Data Presentation Modifications**

Reviewer #1: Line 174: "viral" infection - consider changing the word to pseudovirus

Figure 6A: the labels within the photos blend in with positive staining - consider changing these labels to white.

"Authentic" SARS-CoV-2: It is appreciated that the authors are showing the difference between pseudovirus and whole virus experiments, but "Authentic" isn't really the nomenclature to use. Perhaps just use the word "pandemic"?

Given the advent of evolution of variants of concern (VOCs) which have many mutations in the spike protein with somewhat altered receptor binding affinity/specificity and dependency on different proteases for Spike cleavage, it would be good to indicate that the isolate used by this study matches the earlier or original 'outbreak' virus, versus the emerged VOCs.

Reviewer #2: -Line 76, “Influenza” should not be capitalized

-The term “organoids” does not seem appropriate when referring to human airway cultures on transwells.

-What does the “N” distinction in Fig 1A refer to? Nasal?

-Line 117, Fig 1F should be Fig 1E

-The language is confusion at times around the activity of StcE. For example, in some places it seems the enzyme is removing the entire extracellular domain (e.g. line 125) vs. the associated glycans (e.g. line 129). This should be clarified throughout.

-Fig 3E states “authentic SARS-CoV-2” yet in the text at line 185, this seems to refer to pseudoparticles, and there is no Fig 3F as referred to on line 192.

-Line 255, do you mean Fig 6E instead of Fig 4E?

-Line 302 should read “initial infection”

-please check the y-axis in Figure S2C as the values seem incorrect.

Reviewer #3: (No Response)

PLOS authors have the option to publish the peer review history of their article (what does this mean?). If published, this will include your full peer review and any attached files.

Reviewer #1: **Yes: **Julie McAuley

Reviewer #2: No

Reviewer #3: No
---

## [Decision Letter · Decision Letter 1]

6 Jul 2023

Dear Dr. Strijbis,

Thank you very much for submitting your manuscript "Glycosylated extracellular mucin domains protect against SARS-CoV-2 infection at the respiratory surface" for consideration at PLOS Pathogens. As with all papers reviewed by the journal, your manuscript was reviewed by members of the editorial board and by several independent reviewers. The reviewers appreciated the attention to an important topic. Based on the reviews, we are likely to accept this manuscript for publication, providing that you modify the manuscript according to the review recommendations.

Thank you for the resubmission. Below you can see that 2 out of the 3 reviewers are satisfied with the new version however 1 reviewer has comments remaining. The reviewer is asking for clarification in the text and better quantitation on a couple figures. This can be done without additional experiments, in my opinion. Please let us know if there are any questions about the remaining review.

Sincerely,

Matthew B. Frieman

Guest Editor

PLOS Pathogens

Sara Cherry

Section Editor

PLOS Pathogens

Kasturi Haldar

Editor-in-Chief

PLOS Pathogens

orcid.org/0000-0001-5065-158X

Michael Malim

Editor-in-Chief

PLOS Pathogens

orcid.org/0000-0002-7699-2064

Thank you for the resubmission. Below you can see that 2 out of the 3 reviewers are satisfied with the new version however 1 reviewer has comments remaining. The reviewer is asking for clarification in the text and better quantitation on a couple figures. This can be done without additional experiments, in my opinion. Please let us know if there are any questions about the remaining review.

Reviewer Comments (if any, and for reference):

Reviewer's Responses to Questions

**Part I - Summary**

Reviewer #1: Understanding host-pathogen interactions at the cellular surface are essential for improving our knowledge surrounding infection processes and host-defences. This expert team asks the important question as to whether the cell-surface and gel-forming mucins of the respiratory mucosa aid or hinder the ability for SARS-CoV-2 to infect cells. Using appropriate mucinases, fucosinases, neuroaminidases, and antibody controls the authors conclusively show that the cell surface MUC1 extracellular domain provided a barrier-type role and aided in limiting SARS-CoV-2 infection of the underlying cell. When MUC1 was expressed in it’s natural form, less SARS-CoV-2 was able to gain access to ACE2 (the virus receptor), while removal of the extracellular domain of MUC1 enhanced ACE2 access and infection. The authors also correlate their findings with human ALI organoid cultures, proving biological relevance. The single-cell RNA seq data visualized in diffusion maps matched with the experimental data, where the expected expression of each mucin subtype was shown to be distributed as expected in the ALI and Calu3 cell types. The discussion places the findings of the paper in relevant context – where care is taken to explain that while MUC1 may provide a novel therapeutic target to limit SARS-CoV-2 infection, treatments that alter the homeostasis of this glycoprotein during the course of infection may alter other important functions that MUC1 plays in immune regulatory responses to the infection, but clearly this cs-mucin plays an important role in COVID severity.

Reviewer #2: The authors have added new experiments and addressed many of the concerns raised during initial review. Beyond characterizing mucin expression and the impact of StcE treatment in Calu3 and airway organoid models, the authors show enhanced pseudoparticle and authentic virus infection after StcE treatment in these systems. They have also added new data, including spike binding +/-StcE and proximity ligation assays to bolster their model that tethered mucins block SARS-CoV-2 receptor access.

Still, the staining in Figure 6 (spike binding in airway epithelial organoid cultures) are not very convincing, and the lack of quantification in either Fig 6C or Fig 7 limits the strength of these data. Further, (as noted previously), many aspects of this work have been previously reported in the literature (e.g. PMID: 35879412; PMID: 22929560 etc.) and the mucin-specific impacts remain unresolved here. Thus, overall, the current manuscript adds to a growing body of evidence that tethered mucins are important barrier molecules during SARS-CoV-2 infection, but only moderately advance the field in this area.

Reviewer #3: This resubmission includes new findings that further validate the hypothesis that mucins impede SARS-CoV-2 entry. The investigators have addressed reviewers concerns with new results and with appropriate text additions and modifications. Together with related publications that have evaluated the role of mucins in SARS-CoV-2 infections, this paper strengthens the central hypothesis that mucins exert antiviral effects and this brings more clarity to the field. The works were carefully and competently completed.

**Part II – Major Issues: Key Experiments Required for Acceptance**

Reviewer #1: No major issues identified

Reviewer #2: Line 252 – the StcE- and E447D-treated conditions in Fig 6C look very similar, and only one image of each condition / donor is shown. Quantification here across multiple experiments/images is needed here to support the conclusions.

Figure 7A/B – what is the frequency of ACE2/MUC1 double positive cells? This again seems important to quantify to support their model.

Reviewer #3: (No Response)

**Part III – Minor Issues: Editorial and Data Presentation Modifications**

Reviewer #1: All minor issues identified in my earlier review have been fixed.

Reviewer #2: The authors note that MUC4 and MUC5AC could not be stained in Fig 1D (line 115), indicating these mucins are intracellular. Given that MUC5AC is known to be a secreted mucin and in light of previous data by other groups showing MUC4 expression on the cell surface in airway cultures (PMID: 22929560), a cautionary statement should be included here regarding the limitations of antibodies used, staining protocol, or expression levels of these targets in these cells. As it reads currently, it appears that the authors believe these mucins are simply not expressed on the surface or secreted in this model which is unlikely.

Please note in the methods what controls were used in IF experiments (e.g. secondary antibody only?) to set the thresholds.

Line 188. Figure 3E appears to be authentic virus in the figure, not pseudoparticles as noted here. There is also no figure 3F as referred to on line 196.

Line 331 should read “while ciliated cells”

Line 337 states that MUC1 was “found to be highly expressed on microvilli of basal epithelial cells.” I believe this should state that MUC1 was found on microvilli AND on basal epithelial cells (as demonstrated in Figure 5 of the referenced publication).

Reviewer #3: (No Response)

PLOS authors have the option to publish the peer review history of their article (what does this mean?). If published, this will include your full peer review and any attached files.

Reviewer #1: **Yes: **Julie McAuley

Reviewer #2: No

Reviewer #3: No

Figure Files:

Data Requirements:

Reproducibility:

References:

---

## [Editor Report · Decision Letter 2]

21 Jul 2023

Dear Dr. Strijbis,

We are pleased to inform you that your manuscript 'Glycosylated extracellular mucin domains protect against SARS-CoV-2 infection at the respiratory surface' has been provisionally accepted for publication in PLOS Pathogens.

Best regards,

Matthew B. Frieman

Guest Editor

PLOS Pathogens

Sara Cherry

Section Editor

PLOS Pathogens

Kasturi Haldar

Editor-in-Chief

PLOS Pathogens

orcid.org/0000-0001-5065-158X

Michael Malim

Editor-in-Chief

PLOS Pathogens

orcid.org/0000-0002-7699-2064

Thank you for the resubmission and answering all of the reviewers comments.
---

## [Editor Report · Acceptance letter]

7 Aug 2023

Dear Dr. Strijbis,

We are delighted to inform you that your manuscript, "Glycosylated extracellular mucin domains protect against SARS-CoV-2 infection at the respiratory surface," has been formally accepted for publication in PLOS Pathogens.

Best regards,

Kasturi Haldar

Editor-in-Chief

PLOS Pathogens

orcid.org/0000-0001-5065-158X

Michael Malim

Editor-in-Chief

PLOS Pathogens

orcid.org/0000-0002-7699-2064